# Serious Neurological Adverse Events of Ceftriaxone

**DOI:** 10.3390/antibiotics10050540

**Published:** 2021-05-06

**Authors:** Clémence Lacroix, Annie-Pierre Bera-Jonville, François Montastruc, Lionel Velly, Joëlle Micallef, Romain Guilhaumou

**Affiliations:** 1Centre Régional de Pharmacovigilance, Service de Pharmacologie Clinique, APHM, INSERM, Institut Neurosciences Système, UMR 1106, Aix Marseille Université, 13005 Marseille, France; clemence.lacroix@ap-hm.fr (C.L.); joelle.micallef@ap-hm.fr (J.M.); 2Centre Régional de Pharmacovigilance et d’Information sur le Médicament Centre-Val-de-Loire, Service de Pharmacosurveillance, Centre Hospitalier Régional Universitaire de Tours, 37000 Tours, France; jonville-bera@chu-tours.fr; 3Service de Pharmacologie Médicale et Clinique, Centre de Pharmacovigilance, Pharmacoépidémiologie et d’Informations sur le Médicament, Centre Hospitalier Universitaire, Faculté de Médecine, 31000 Toulouse, France; francois.montastruc@univ-tlse3.fr; 4Unité Clinique de Pharmacologie Psychiatrique, Faculté de Médecine, Centre Hospitalier Universitaire, 31000 Toulouse, France; 5Department of Anaesthesiology and Critical Care Medicine, University Hospital Timone, Aix Marseille Université, 13005 Marseille, France; lionel.velly@ap-hm.fr; 6CNRS, INT, Institut Neurosci Timone, UMR 7289, Aix Marseille Université, 13005 Marseille, France; 7Laboratoire de Pharmacologie Clinique, Service de Pharmacologie Clinique, APHM, INSERM, Institut Neurosciences Système, UMR 1106, Aix Marseille Université, 13005 Marseille, France

**Keywords:** antibiotics, ceftriaxone, neurotoxicity, central nervous system, neurologic, adverse effects, pharmacovigilance

## Abstract

We described ceftriaxone-induced CNS adverse events through the largest case series of Adverse Drug Reactions (ADRs) reports, from 1995 to 2017, using the French Pharmacovigilance Database. In total, 152 cases of serious CNS ADRs were analyzed; 112 patients were hospitalized or had a prolonged hospitalization (73.7%), 12 dead (7.9%) and 16 exhibited life-threatening ADRs (10.5%). The median age was 74.5 years, mainly women (55.3%), with a median creatinine clearance of 35 mL/min. Patients mainly exhibited convulsions, status epilepticus, myoclonia (*n* = 75, 49.3%), encephalopathy (*n* = 45, 29.6%), confused state (*n* = 34, 22.4%) and hallucinations (*n* = 16, 10.5%). The median time of onset was 4 days, and the median duration was 4.5 days. The mean daily dose was 1.7 g mainly through an intravenous route (*n* = 106, 69.7%), and three patients received doses above maximal dose of Summary of Product Characteristics. Ceftriaxone plasma concentrations were recorded for 19 patients (12.5%), and 8 were above the toxicity threshold. Electroencephalograms (EEG) performed for 32.9% of the patients (*n* = 50) were abnormal for 74% (*n* = 37). We described the world’s biggest case series of ceftriaxone-induced serious CNS ADRs. Explorations (plasma concentrations, EEG) are contributive to confirm the ceftriaxone toxicity-induced. Clinicians may be cautious with the use of ceftriaxone, especially in the older age or renal impairment population.

## 1. Introduction

Common prescriptions can lead to uncommon reactions, such as Central Nervous System (CNS) Adverse Drug Reactions (ADRs) [1]. Antibiotics are still an under-recognized etiology compared to their high consumption in hospitals; about 50% of hospitalized patients receive at least one antibiotic during their hospital stay [2]. CNS ADRs are reported with a frequency of less than 1% with antimicrobial agents, although signs of encephalopathy, such as delirium, altered mental status or even convulsions, frequently occur during hospitalizations [3,4]. As antibiotics are more and more used to fight against a large number of new resistant bacteria, it is important to better know and prevent ADRs of these medications, and particularly CNS ones [5,6,7,8,9]. Clinicians do not know much about the relationship between neurotoxicity and antibiotics, although CNS ADRs can lead to life-threatening issues and even death. This neurotoxicity could often be prevented by a dose adjustment to renal function. However, unlike other cephalosporins characterized by hydrophilic properties with extensive renal clearance and short elimination half-life, ceftriaxone presents an uncommon pharmacokinetics profile. Indeed, it shows an extended albumin binding, mixed biliary and renal clearance and extended elimination half-life, explaining once-daily standard dosing regimen [10,11]. To our knowledge and based on literature, encephalopathy is not a well-known ADR of ceftriaxone with only around twenty case reports published in the literature [12,13,14,15,16,17,18,19,20,21,22,23,24]. Due to the scarcity of evidence in the literature and recent work by our team on a large pharmacovigilance database showing seven times as many serious cases reports of CNS ADRs involving ceftriaxone than in the literature [25], this work’s aim is to characterize accurately ceftriaxone-induced serious CNS ADRs from clinical and pharmacokinetic points of view.

## 2. Results

### 2.1. General Characteristics of Serious Case Reports

We identified a total of 152 serious reports in the French Pharmacovigilance Database describing 216 CNS ADRs analyzed in our study. Among these reports, 145 (95.4%) were reported by hospital clinicians. Reports involving women accounted for 55.3% (*n* = 84). The median age was 74.5 (Q1:63; Q3:84.3), and 69.7% (*n* = 106) of the reports concerned patients over 65 years old. The ADR led to a hospitalization or prolonged hospitalization in 73.7% (*n* = 112), while deaths and life-threatening ADRs were reported in, respectively, 7.9% (*n* = 12) and 10.5% (*n* = 16).

Creatinine clearance was registered for 59 reports (38.8%) with a median of 35 mL/min (Q1:20; Q3:59.5). Among these 59 creatinine clearances, we observed 74.6% (*n* = 44) with renal impairment (Table 1).

### 2.2. Administration Route, Daily Dose and Indications

Ceftriaxone was administered intravenously in 106 reports (69.3%), through intramuscular injections in 16 reports (10.5%) and sub-cutaneous injections in 14 reports (9.2%). The mean daily dose was 1.7 g (0.5–6) and for 3 patients the daily dose was over the maximum dose in the Summary of Products Characteristics (SmPC) (i.e., 4 g). Two of them were treated with 6 g daily for meningitis (weights 44 kg; 54 kg). Figure 1 shows the different indications of ceftriaxone. 

### 2.3. Concomitant Administration of Antibiotics

We found a concomitant administration of other antibiotics for 72 patients (47.4%), and 16 patients were treated by 3 or more antibiotics. We mostly observed a concomitant administration of fluoroquinolones (*n* = 34), metronidazole (*n* = 24) and other beta-lactams (*n* = 14).

### 2.4. Type of Serious ADRs Reported

A total of 216 serious CNS ADRs was reported. The most frequently reported serious neurologic ADRs were mainly encephalopathy (*n* = 45, 20.8%), convulsions (without other information) (*n* = 28, 13%), myoclonia (*n* = 13, 6%), status epilepticus (*n* = 11, 5.1%), tonic-clonic seizures (*n* = 4), tonic seizures (*n* = 2), partial seizures (*n* = 2), clonic seizures (*n* = 1) and focal seizures (*n* = 1). 

The main psychiatric ADRs reported included a confused state (*n* = 34, 15.7%) and hallucinations (*n* = 16, 7.4%).

The median time of onset of CNS ADRs was 4 days (Q1:2; Q3:9) after starting ceftriaxone. The median duration of the clinical manifestations was 4.5 days (Q1:2; Q3:7). 

### 2.5. Explorations

#### 2.5.1. Plasma Concentrations

Ceftriaxone plasma concentrations were available for 19 patients (12.5%), and 8 were over the toxicity level (>100 µg/mL). Figure 2 summarizes results of plasma concentrations.

#### 2.5.2. Electroencephalograms

The results of an electroencephalogram were available in 50 reports (32.9%) and were abnormal for 37 (74%). The electroencephalogram’s specific findings were described in 18 reports (48.6%), including mainly slow waves discharges, following by triphasic waves, delta-theta activity or biphasic waves.

## 3. Discussion

Our study shows the largest case series of serious CNS ADRs reports with ceftriaxone, with seven times more reports recorded in our work than published in the literature (152 vs. 21). Cases of ceftriaxone-induced serious CNS ADRs were reported from 1995 to 2017, with a majority of hospitalizations and life-threatening ADRs and even 12 deaths. Patients were mainly over 65 years old and experienced renal impairment (median creatinine clearance: 35 mL/min). They exhibited CNS manifestations after a median time of 4 days to onset with mainly convulsions and encephalopathy, as well as confused state and hallucination. 

Ceftriaxone is used both in the hospital and in the community sector, with high consumption in France [26] and in the world [27]. This cephalosporin is widely prescribed by clinicians thanks to its antimicrobial broad-spectrum and its pharmacokinetics specificities. It is a very lipophilic molecule with a mixt elimination and a long elimination half-life allowing only one daily administration [28]. As a result, ceftriaxone apparently does not require a dose adjustment to the renal function [29], unlike other beta-lactams [30]. However, our results show that patients over 65 years and/or suffering from a renal impairment are at risk of developing serious CNS ADRs with this molecule as in previous studies [8,31] and in particular in our recent study including all cephalosporins [25]. Indeed, age can be related to pharmacodynamic changes through a higher serious CNS ADRs predisposition and pharmacokinetics ones with a decrease in protein binding. The unbound fraction of ceftriaxone is therefore increased in the elderly because of a decrease in albumin levels. Renal impairment also affects the elimination of the unbound fraction of the molecule and is usually related to age [32,33,34]. Patients can exhibit renal impairment at baseline, but they can also develop it acutely during treatment due to their medical conditions. A longer elimination then implies an accumulation of ceftriaxone and results in clinical CNS manifestations. Indeed, serum creatinine and estimated glomerular filtration rate were observed as significant covariates of ceftriaxone in recent population pharmacokinetic studies [35,36,37,38,39]. These results are also in accordance with recent pharmacokinetic studies that conclude that ceftriaxone daily dose has to be adjusted to renal function and weight to avoid toxicity plasma concentrations in patients exhibiting meningitis [35]. Another hypothesis linked to renal impairment is an accumulation of toxic organic acids or an alteration of pH contributing to impair active transport of the molecule from cerebrospinal fluid to blood, promoting neurotoxicity [40]. Excessive dosage is also considered as a risk factor. Nevertheless, our results show only 3 patients who received daily doses over the SmPC with two hospitalized for meningitis. A recent study highlights a well tolerability of high-dose ceftriaxone in CNS infections with a particular caution in patients with advanced age or renal impairment [41].

Regarding published literature, only 21 cases of ceftriaxone-induced neurotoxicity have been reported and only two without renal impairment (Table 2), which is surprising considering our large series of reports (152 vs. 21). Serious CNS ADRs are widely represented by neurologic effects with mainly convulsions, myoclonia and signs of encephalopathy. The pathophysiological mechanism of such neurotoxicity is not totally elucidated, but thought to be similar to other cephalosporins with an increase in glutamatergic excitation, a decrease in γ-amino-butyric acid (GABA) via the competitive concentration-dependent inhibition in the GABA-A receptor complex [42], a release of endotoxins and cytokines, and an increase in excitatory ability related to N-methyl-d-aspartate receptors and α-amino-3-hydroxy-5-methyl-isoxazolepropionate receptors [5]. Interestingly, ceftriaxone also interacts with the glutamate transporter GLT-1 and may inhibit the development of opioid-induced hyperalgesia [43] or attenuate drug-seeking behavior with alcohol or cocaine in rats as a result [44,45,46]. Our study is in line with the few cases reported in Medline. Indeed, patients mostly exhibited encephalopathy and then other CNS ADRs, such as convulsion and myoclonia or even confused state and hallucination. As the French Pharmacovigilance Database uses MedDRA coding, the distinction between all signs of encephalopathy is done. Nevertheless, for example, when patients exhibited only convulsions, it is only coded “convulsion.” Serious CNS ADRs associated with ceftriaxone use are about the same compared to the study with all cephalosporins [25]. Although SmPC of this drug only mentions “convulsion” as a potential ADR with an indefinite frequency [29], clinicians should be aware of the risk of encephalopathy and other neurological disorders considering our results. To evaluate these serious CNS ADRs correctly, several explorations are important to guide to a prompt and accurate diagnosis. Plasma concentrations have been recorded for 19 patients with 8 over the toxicity threshold [41]. However, these data remained difficult to interpret owing to the lack of information about sample time or treatment discontinuation. Plasma concentration data are not always mentioned in literature case reports. Nevertheless, exploration like plasma concentration is contributive to confirm the drug toxicity and describes drug exposure. Furthermore, in specific populations, such as critical care patients, therapeutic drug monitoring is now recommended to optimize beta-lactam antibiotics treatment and avoid neurotoxicity [47,48]. EEG is also a precious tool to investigate CNS ADRs, ¾ of the EEG in this study were abnormal when recorded, with signs of encephalopathy. Previous studies highlighted these specific signs with cephalosporin-induced encephalopathy, such as triphasic waves or slow-wave discharges [49,50]. Plasma concentrations and EEG provide essential cues regarding the diagnosis. In several case reports, patients did not exhibit clinical manifestations but only biological ones with electroencephalographic features such as non-convulsive status epilepticus [12,13]. Meningitis could lead to CNS manifestations, like other severe infectious diseases (septic shock, for example). It could be confusing for the assessment of ceftriaxone. Similarly, antibiotics such as other beta-lactams, quinolones or metronidazole are known to provide CNS ADRs [3,5] and thus could also lead to a difficult assessment of ceftriaxone. After eliminating clinical etiologies, clinicians then suspect iatrogenic etiologies. Each case reported to the pharmacovigilance center undergoes a pharmacological, clinical and biological assessment by pharmacologists. Explorations like EEG and plasma concentrations are all the more important in this context.

Several publications of safety communications in the US by the Food and Drug Administration (FDA) in 2012 [51] and in France by the Agence Nationale de Sécurité du Médicament et des produits de santé (ANSM) in 2014 [52] and again in 2018 [53] were published concerning cefepime. Conversely, ceftriaxone was defined as a low risk of neurotoxicity agent compared to other antimicrobials in 2011 [8]. However, owing to the significant number of patients treated with this antibiotic per year, serious neurotoxicity events have to be considered, especially in some specific populations. Advanced age or renal-impaired patients have been described as a high-risk population, and healthcare professionals have to be alerted in order to optimize ceftriaxone treatment in these vulnerable populations. Moreover, as these ADRs are poorly known, they are probably underdiagnosed and thus underreported. Indeed, underreporting is the main limitation of our study, a well-known weakness in spontaneous reporting systems. The information is neither always complete nor ho-mogeneous and several biases have been well described: underreporting, missing data, and absence of the total number of treated patients [54]. As previously discussed in the manuscript, another limitation consists in the report of concomitant administration of antibiotics known to provide CNS ADRs. Nevertheless, the French pharmacovigilance database allows to collect standardized information from spontaneous reports, previ-ously validated one by one by trained and experimented pharmacologists. Despite their limitations, this spontaneous reporting system of ADRs remains one of the most useful tool to generate signals in drug safety and post-marketing surveillance, as high-lighted by the largest-to-date CNS ADRS associated with ceftriaxone.

## 4. Materials and Methods

### 4.1. Data Set

The French Pharmacovigilance Database is implemented by the French pharmacovigilance network made of 31 regional centers located in French University hospitals. Healthcare professionals have to report all suspected adverse drug reactions (ADRs), especially if serious and/or unexpected, to the French Medicine Agency (Agence Nationalede Sécurité du Médicament et des produits de santé (ANSM)) via a regional pharmacovigilance center. Since 2011, all patients can also declare ADRs on a voluntary basis. This system is integrated into a European organization for drug monitoring and authorization. Reports are validated by clinical pharmacologists in the relevant regional pharmacovigilance center (according to the French drug causality method of imputability [55]) before being recorded in the French pharmacovigilance database. This database is administered by the French Medicine Agency, the Agence Nationale de Sécurité du Médicament et des Produits de Santé (ANSM) and updated daily by practitioners in the French regional pharmacovigilance center network with spontaneously reported or collected cases, in accordance with good pharmacovigilance practice. Each year, around 40,000 ADR reports are collected by this system. Each ADR recorded includes, if possible, information regarding the patient’s demographics and medical history, the suspected drug exposure and causality (dose, formulation, indication, challenge, withdrawal, rechallenge and causality score), concomitant medications, ADR severity (nonserious or serious according to the World Health Organization terminology), and the ADR outcome. Data concerning ADRs and indications of use are coded using the Medical Dictionary for Regulatory Activities (MedDRA) classification, which is organized into five hierarchical levels, “System organ class” (SOC), “High-Level Groups Terms” (HLGTs), “High-Level Terms” (HLTs), “Preferred Terms” (PTs) and “Lowest Level Terms” (LLTs). All data are anonymized. 

### 4.2. Population and Cases

We performed a retrospective, descriptive analysis. We reviewed all spontaneous notifications of serious CNS ADRs recorded with ceftriaxone (Anatomical Therapeutic Chemical: J01DD04) exposure from 1995 (the first case reported) to the 31st December of 2017 (end of the study period). We selected CNS ADRs using the MedDRA dictionary specific codes, according to the SOC’s “Nervous system disorders” and “Psychiatric disorders.” We also collected general information concerning patients’ characteristics. Administration routes, the daily dose (at the CNS ADR onset), concomitant use of other antibiotics and indications of ceftriaxone therapy were also collected. Concerning the types of CNS ADRs, we used Preferred Terms (PTs) to characterize serious CNS ADRs. We then gathered PTs when medically closed. We collected the results of exams usually performed in case of encephalopathy: electroencephalograms and plasma concentrations during the CNS ADR, in the notifications verbatim. As the data were provided by a pharmacovigilance database, some information was missing. 

### 4.3. Statistical Analysis 

We described general characteristics of serious case reports, as well as administration route, daily doses, concomitant use of other antibiotics, and indications. We also performed a description of the type of serious ADRs. We compared ceftriaxone plasma concentrations to target range (20–100 µg/mL) [47]. We estimated the creatinine clearance by the Cockroft–Gault formula, and we defined renal impairment by creatinine clearance under 60 mL/min. Finally, we described results of encephalograms if available. We described quantitative variables in terms of median and interquartile (Q1, Q3) and qualitative variables as proportion. We performed all analysis and visualization using Excel^®^ and R version 4.0.1.

### 4.4. Data Availability Statement

The data that support the findings of this study are available on request from the corresponding author upon reasonable request. The data are not publicly available due to privacy regulations.

As this study was performed retrospectively using routinely collected anonymous data, it did not require any ethics committee approval, which is in line with French regulations for mandatory reporting of pharmacovigilance cases by healthcare professionals and patients.

## 5. Conclusions

This study, which is the world’s biggest case series, shows that ceftriaxone can lead to life-threatening conditions with serious CNS ADRs and even death. The early recognition followed by discontinuation and/or a switch of treatment usually allows rapid recovery. That is why healthcare professionals have to be particularly cautious with the use of ceftriaxone, especially for patients with altered creatinine clearance.

## Figures and Tables

**Figure 1 antibiotics-10-00540-f001:**
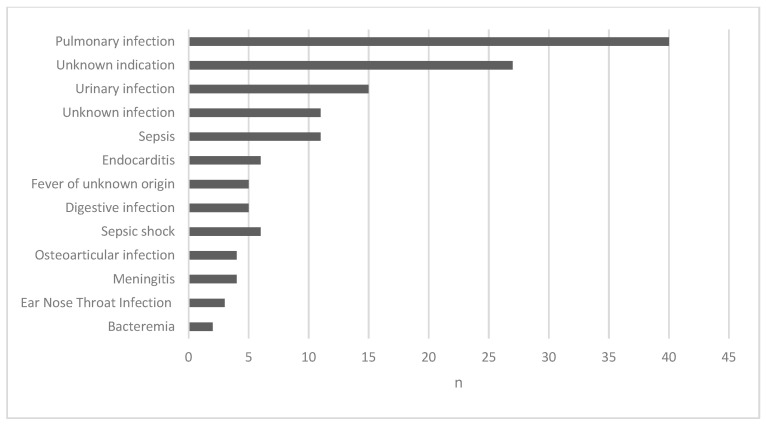
Number of serious reports of CNS ADRs related to ceftriaxone per indications.

**Figure 2 antibiotics-10-00540-f002:**
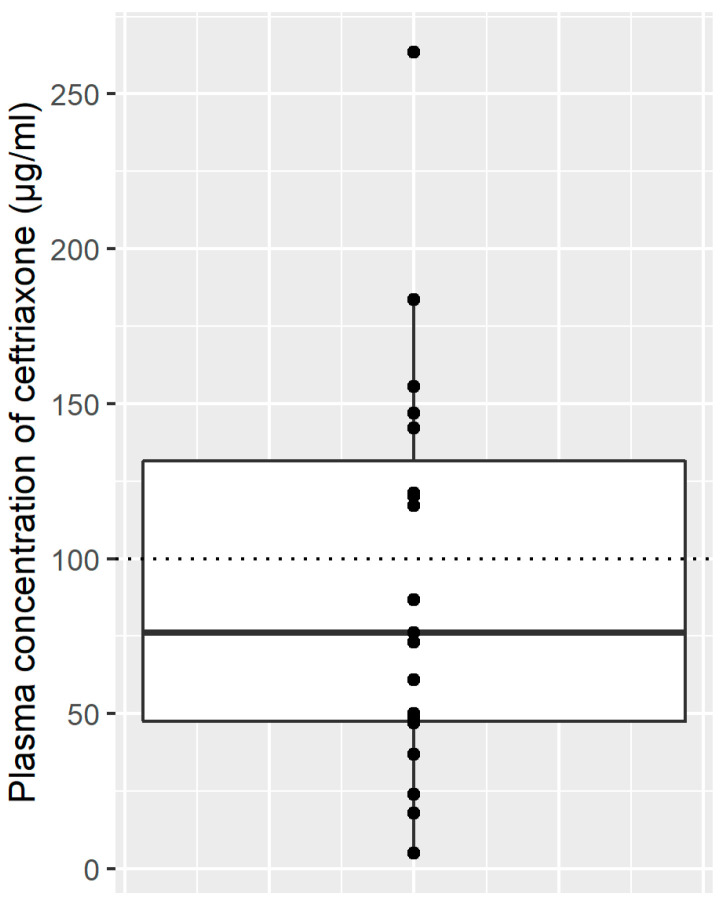
Plasma concentrations of ceftriaxone (µg/mL) are represented by points. The box plot represents the first and third quartiles (Q1–Q3) of plasma concentrations of ceftriaxone; the black horizontal line represents the median of plasma concentrations of ceftriaxone; the dotted line represents the toxicity level of ceftriaxone (100 µg/mL).

**Table 1 antibiotics-10-00540-t001:** Demographic and clinical characteristics of patients.

Patients’ Characteristics	
Female; *n* (%)	84 (55.3)
Age (years); median (Q1; Q3)	74.5 (63; 84.3)
Age > 65 years old; *n* (%)	106 (69.7)
Creatinine clearance (mL/min); median (Q1; Q3)	35 (20; 59.5)

mL milliliter, min minute, Q quartile.

**Table 2 antibiotics-10-00540-t002:** Systematic literature review of neurotoxicity attributable to ceftriaxone.

Patient	Age	Sex	Renal Function	Dose (g/Day)	Through Concentration (µg/mL)	Indication	Neurological Manifestations	Electroencephalogram Findings	Days to Onset	Days to Remission	Treatment	Ref
**1**	83	F	CKD	2	-	Pneumonia	Drowsiness, myoclonus	-	4	5	Discontinue, AED	[12]
**2**	78	F	CKD	2	-	Meningitis	Drowsiness, myoclonus	-	6	5	Discontinue, AED	[12]
**3**	12	F	CKD	100 mg/kg	-	Sepsis	Confusion, visual hallucinations, facial myoclonus (rechallenge +)	Bursts and runs of generalized spike and spike wave discharges	3	2	Discontinue, AED	[14]
**4**	60	F	ARF	2	-	Hypogastric pain, fever	Altered mental status, apathy, somnolence	Periodic generalized triphasic waves	4	2	Discontinue	[15]
**5**	65	F	CKD	2	-	Chill, fever	Altered mental status, generalized myoclonic jerks	Generalized slowing with superimposed almost continuous or periodic bursts of sharp waves or sharp and slow wave activity	5	2	Discontinue	[16]
**6**	8	M	Normal	1	-	Diarrhea, fever	Altered mental status, apathy, somnolence	-	3	3	Discontinue	[17]
**7**	37	F	CKD, PD	2	-	Peritonitis	Agitation, paranoia, visual hallucinations	Moderate diffuse nonspecific slowing without epileptogenic activity	3	1.5	Discontinue	[18]
**8**	24	F	CKD	2	-	Recurrent urinary tract infection	Confusion after general tonico-clonic seizure	Continuous rhythmic generalized 2 to 3 Hz sharp- wave activity, extensive epileptiform activity	3	3	Discontinue, AED	[13]
**9**	71	M	CKD	2	-	Wound infection	Meaningless speech, inability to walk, sleepiness	Diffuse slow- wave activity	5	5	Discontinue, AED	[13]
**10**	56	M	CKD, HD	4 (days 1–7) 2 (days 8–15)	-	Sepsis	Altered mental status, facial myoclonus, sporadic phonation	Bursts of generalized, high-voltage slow-wave activity	7	5	Discontinue	[19]
**11**	72	M	CKD	4 (days 1–7) 2 (days 8–10)	472 (day 8) 173 (day 10)	Pneumonia	Altered mental status, spasms of legs	Diffuse slow-wave activity	8	6	Discontinue	[20]
**12**	75	F	CKD	2	304 (day 4) 331 (day 6) 422 (day 9)	Diverticulitis	Agitation, hyperkinesia, confusion	Slow-wave activity	9	4	Discontinue	[20]
**13**	68	F	CKD	4 (days 1–7) 2 (days 8–23)	172 (day 2) 178 (day 4) 188 (day 7)	Pyogenic arthritis	-	-	-	-	-	[20]
**14**	76	M	RI	4	-	Endocarditis	Agitation, confusion, coma	Triphasic waves	14	2	Discontinue	[21]
**15**	70	F	ARF	4	-	Meningitis	Encephalopathy, myoclonus	Severe slowing triphasic waves	3	-	-	[22]
**16**	80	F	Normal	2.5	-	Pneumonia	Encephalopathy	Moderate slowing triphasic waves	2	-	-	[22]
**17**	80	F	HD	4	-	Cellulitis	Choreoathetosis	-	5	12	Discontinue	[23]
**18**	72	F	HD	1	-	Catheter-related infection	Choreoathetosis	-	2	1	Discontinue	[23]
**19**	76	M	HD	2	-	Pneumonia	Choreoathetosis	-	6	-	Discontinue	[23]
**20**	76	M	HD	2	-	Catheter-related infection	Choreoathetosis	-	5	2	Discontinue	[23]
**21**	86	F	HD	1 (days 1–3) 2 (days 3–13)	130 (day 9) (LCR 10.2)	H. cinaedi bacteremia	Altered mental status, decreased level of consciousness, myoclonic jerks (right shoulder and arm)	Generalized triphasic waves	13	4	Discontinue	[24]

F: female, M: male, CKD: Chronic Kidney Disease, ARF: Acute Renal Failure, PD: Peritoneal Dialysis, RI: Renal impairment, HD: Hemodialysis, AED: Anti-Epileptic Drugs.

## Data Availability

The data that support the findings of this study are available on request from the corresponding author upon reasonable request. The data are not publicly available due to privacy regulations. As this study was performed retrospectively using routinely collected anonymous data, it did not require any ethics committee approval, which is in line with the French regulations for mandatory reporting of pharmacovigilance cases by healthcare professionals and patients.

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
