# Peer review of "Serious Neurological Adverse Events of Ceftriaxone"

_antibiotics, 2021, doi:10.3390/antibiotics10050540_

Round 1

Reviewer 1 Report

This study is the first to report a large number of CNS ADRs upon ceftriaxone treatment and thus constitutes an important publication for raising awareness about these poorly understood and underdiagnosed ADRs. The authors made use of the excellent French pharmacovigilance database for case extraction and provided evidence for the necessity of ceftriaxone dose adjustment according to renal function.

My suggestions for improving the manuscript at this time are:

that the authors clarify the details of ceftriaxone pharmacokinetics (specific parameters and how it compares to other cefalosporins)

the authors clarify which cases resulted in death? Are these cases corelated with ceftriaxone overdose?

It would be helpful to see details regarding the causality score for these ADRs and also the concomitant use of other medications, especially since in some instances there can be confounding by indication (e.g. meningitis)

Author Response

This study is the first to report a large number of CNS ADRs upon ceftriaxone treatment and thus constitutes an important publication for raising awareness about these poorly understood and underdiagnosed ADRs. The authors made use of the excellent French pharmacovigilance database for case extraction and provided evidence for the necessity of ceftriaxone dose adjustment according to renal function.

We are very grateful to the reviewer for the positive feedback. We thoroughly agree with the comments of the reviewer, and we updated our manuscript accordingly. We have also made an extensive review of English language and style.

My suggestions for improving the manuscript at this time are:

that the authors clarify the details of ceftriaxone pharmacokinetics (specific parameters and how it compares to other cefalosporins)

We agree with the referee and data on ceftriaxone pharmacokinetics have been added in the introduction section: “Unlike other cephalosporins characterized by hydrophilic properties with extensive renal clearance and short elimination half-life, ceftriaxone presents an uncommon pharmacokinetics profile. Indeed, it shows an extended albumin binding, mixed biliary and renal clearance and extended elimination half-life, explaining once-daily standard dosing regimen [10,11]” (Patel and al, 1981; Joynt and al, 2001) (lines 52-58).

the authors clarify which cases resulted in death? Are these cases correlated with ceftriaxone overdose?

Our study shows 152 serious CNS ADRs reports with the use of ceftriaxone. Among these reports, 12 have led to deaths (see section 2.1) after exhibiting encephalopathy (n=5), convulsions (n=4) and coma (n=3). Among these deaths, there are no documented reports related to ceftriaxone overdose. Indeed, no interpretable ceftriaxone plasma concentrations have been found (sample time missing, delayed sampling). Noteworthy, available data concerning EEG show abnormalities in 3 reports.

As these ADRs of ceftriaxone are poorly known, it is all the more important to manage explorations such as plasma concentrations and EEG to guide to a prompt diagnosis and avoid severe consequences. In the discussion section, several sentences support this comment: 

To evaluate these serious CNS ADRs correctly, several explorations are important to guide to a prompt and accurate diagnosis”; “exploration like plasma concentration is contributive to confirm the drug toxicity and describes drug exposure”; “Plasma concentrations and EEG provide essential cues regarding the diagnosis”.    

It would be helpful to see details regarding the causality score for these ADRs and also the concomitant use of other medications, especially since in some instances there can be confounding by indication (e.g. meningitis)

We agree with the referee, other medications could provide CNS ADRs and thus could be confounding factors. Clinicians in pharmacovigilance centers provide a pharmacological, biological and chronological assessment and the use of other medications are always taken into account. This is part of the causality score. Indeed the French method for causality assessment in pharmacovigilance is based on the evaluation of 3 criteria: chronology, semiology and bibliographic data. The score provided by this algorithmic method makes sense for individual case reports. In our study, we present aggregated data from 152 serious case reports of CNS ADRs with the use of ceftriaxone. It is difficult to make an advised interpretation of a global causality score. Nevertheless, we present parts of this causality score like the chronology (see 2.4 in results section). Based on the medical data of the 152 serious reports, the median time of CNS onset was 4 days after starting ceftriaxone and the median duration was 4.5 days. These delays are in line with data available in literature (mainly case reports, presented in table 2 of the manuscript). We also present semiological features used in the causality score through results of explorations (EEG, plasma concentrations) (see 2.5 in results section). To better characterize our case reports and to answer the referee, we have added results of concomitant use of other antibiotics. Indeed other antibiotics such as other betalactams, metronidazole, or quinolones can provide CNS ADRs (Bhattacharyya and al, 2016; Deshayes and al, 2017). In conclusion, the aggregated causality score was not added as it is allocated individually at a given moment, on the basis of available information and may thus change over the time. Interestingly, it could change according to pharmacological knowledge (awareness of a potential problem, mechanisms of the ADR) and clinical knowledge (etiological investigations) (Miremont-Salamé and al, 2016).  

To address your comment, we have added the following information:

-in the Materials and Methods section, population and cases: “Administration routes, daily dose (at the CNS ADR onset), concomitant use of other antibiotics and indications of ceftriaxone therapy were also collected.” (line 254)

-in the Materials and Methods section, statistical analysis: “We described general characteristics of serious case reports, as well as administration route, daily doses, concomitant use of other antibiotics, and indications.” (line 263) 

-in the results section: “2.3. Concomitant use of other antibiotics” (lines 95-99)

Regarding indications of use, we agree with the referee that indications could be a confounding factor. Nevertheless, our case series mainly come from hospital clinicians. After eliminating clinical etiologies, clinicians then suspect iatrogenic etiologies. Hence, it is important to raise awareness of clinicians about CNS ADRs of ceftriaxone. As we mentioned in our study, explorations like EEG and plasma concentrations are contributive to lead to a prompt diagnostic.

In the discussion section, several sentences support this comment: 

To evaluate these serious CNS ADRs correctly, several explorations are important to guide to a prompt and accurate diagnosis”; “exploration like plasma concentration is contributive to confirm the drug toxicity and describes drug exposure”; “Plasma concentrations and EEG provide essential cues regarding the diagnosis”.       

We have thus added the following paragraph:

-in the section discussion: “Meningitis could led to CNS manifestations, like other severe infectious diseases (septic shock for example). It could be confusing for the assessment of ceftriaxone. Similarly, antibiotics such as other betalactams, quinolones or metronidazole are known to provide CNS ADRs [3,5] and thus could also led to a difficult assessment of ceftriaxone. After eliminating clinical etiologies, clinicians then suspect iatrogenic etiologies. Each case reported to the pharmacovigilance center undergoes a pharmacological, clinical and biological assessment by pharmacologists. Explorations are all the more important in this context” (lines 201-208).

References

-Bhattacharyya, S.; Darby, R.R.; Raibagkar, P.; Gonzalez Castro, L.N.; Berkowitz, A.L. Antibiotic-Associated Encephalopathy. Neurology 2016, 86, 963–971, doi:10.1212/WNL.0000000000002455.

-Deshayes, S.; Coquerel, A.; Verdon, R. Neurological Adverse Effects Attributable to β-Lactam Antibiotics: A Literature Review. Drug Safety 2017, 40, 1171–1198, doi:10.1007/s40264-017-0578-2.

- Joynt, G.M. The Pharmacokinetics of Once-Daily Dosing of Ceftriaxone in Critically Ill Patients. Journal of Antimicrobial Chemotherapy 2001, 47, 421–429, doi:10.1093/jac/47.4.421.

-Miremont-Salamé, G.; Théophile, H.; Haramburu, F.; Bégaud, B. Causality assessment in pharmacovigilance: The French method and its successive updates. Therapie 2016, 71, 179-186, doi: 10.1016/j.therap.2016.02.010

-Patel, I.H.; Chen, S.; Parsonnet, M.; Hackman, M.R.; Brooks, M.A.; Konikoff, J.; Kaplan, S.A. Pharmacokinetics of Ceftriaxone in Humans. Antimicrob. Agents Chemother. 1981, 20, 634–641, doi:10.1128/aac.20.5.634.

Reviewer 2 Report

A clinically very interesting article describing CNS ADRs related to ceftriaxone administration is presented. The authors report a significantly higher incidence of these adverse reactions compared to the literature.

I think the article is suitable for publication, but the following comments need to be addressed:

  1. In Figure 1, the diagnoses are unclear, what is the difference between septicemia (a term that I personally consider obsolete) and bacteremia? Bacteremia means the presence of bacteria in the blood, which does not necessarily mean infection and may not in all cases be an indication for ceftriaxone.
  2. I recommend adjusting the term unknown fever to a fewer of unknown origin.
  3. If septicemia is meant sepsis, how did the authors distinguish the possible clinical manifestations of this serious diagnosis, where does the confuxional state belong?
  4. The above question is more general, in the case of severe bacterial infections, the CNS ADRs mentioned may occur and it is not necessarily a side effect of ceftriaxone, which is generally considered a safe drug. I consider this comment to be essential and without its solution I cannot recommend the paper for acceptance.
  5. Beta-lactam antibiotics are often combined with other antibiotics, such as aminoglycosides. Did the authors consider the possible neurotoxic effect of these antibiotics when administered with ceftriaxone?
  6. I consider it appropriate to add to the chapter Methods the criteria for adverse reactions that have been applied in the case of association with cefriaxone.

Author Response

A clinically very interesting article describing CNS ADRs related to ceftriaxone administration is presented. The authors report a significantly higher incidence of these adverse reactions compared to the literature.

We are very grateful to the reviewer for the positive feedback. We thoroughly agree with the comments of the reviewer, and we updated our manuscript accordingly.

I think the article is suitable for publication, but the following comments need to be addressed:

1. In Figure 1, the diagnoses are unclear, what is the difference between septicemia (a term that I personally consider obsolete) and bacteremia? Bacteremia means the presence of bacteria in the blood, which does not necessarily mean infection and may not in all cases be an indication for ceftriaxone.

We agree with the referee and have harmonized presentation of our data concerning indications of ceftriaxone use (Figure 1, page 3), according to the Third International Consensus Definitions for Sepsis and Septic Shock criteria (Singer and al, JAMA 2016).

2. I recommend adjusting the term unknown fever to a fewer of unknown origin.

We agree with the referee and the term “unknown fever” in figure 1 has been changed into “fever of unknown origin” (page 3).

3. If septicemia is meant sepsis, how did the authors distinguish the possible clinical manifestations of this serious diagnosis, where does the confuxional state belong?

Regarding indications of use, we agree with the referee that indications could be a confounding factor. Nevertheless, our case series mainly come from hospital clinicians. After eliminating clinical etiologies, clinicians then suspect iatrogenic etiologies. Hence, it is important to raise awareness of clinicians about CNS ADRs of ceftriaxone. As we mentioned in our study, explorations like EEG and plasma concentrations are contributive to lead to a prompt diagnosis.

In the discussion section, several sentences support this comment: 

“To evaluate these serious CNS ADRs correctly, several explorations are important to guide to a prompt and accurate diagnosis”; “exploration like plasma concentration is contributive to confirm the drug toxicity and describes drug exposure”; “Plasma concentrations and EEG provide essential cues regarding the diagnosis”.

We acknowledge this is an interesting addition to the discussion. We have therefore added the following paragraphs:

-in the Material and Methods data set section: “French pharmacovigilance database is implemented by the French pharmacovigilance network made of 31 regional centers located in French University hospitals. Healthcare professionals have to report all suspected adverse drug reactions (ADRs), especially if serious and/or unexpected, to the French Medicine Agency (Agence Nationale de Sécurité du Médicament et des produits de santé (ANSM)) via a regional pharmacovigilance center. Since 2011, all patients can also declare ADRs on a voluntary basis.” (lines 222-227)

-in the discussion section: “Meningitis could lead to CNS manifestations, like other severe infectious diseases (septic shock for example). It could be confusing for the assessment of ceftriaxone. Similarly, antibiotics such as other betalactams, quinolones or metronidazole are known to provide CNS ADRs [3,5] and thus could also lead to a difficult assessment of ceftriaxone. After eliminating clinical etiologies, clinicians suspect iatrogenic etiologies. Each case reported to the pharmacovigilance center undergoes a pharmacological, clinical and biological assessment by pharmacologists. Explorations are all the more important in this context.” (lines 201-208)

4. The above question is more general, in the case of severe bacterial infections, the CNS ADRs mentioned may occur and it is not necessarily a side effect of ceftriaxone, which is generally considered a safe drug. I consider this comment to be essential and without its solution I cannot recommend the paper for acceptance.

In our study, we describe a case series of 152 serious reports of CNS ADRs with the use of ceftriaxone. Reports were mainly notified by hospital clinicians. Clinicians have to report all suspected ADRs to pharmacovigilance center. ADRs are suspected after ruling out clinical etiologies. That is why explorations like EEG and plasma levels are important to guide to a prompt diagnosis.

In the discussion section, several sentences support this comment: 

To evaluate these serious CNS ADRs correctly, several explorations are important to guide to a prompt and accurate diagnosis”; “exploration like plasma concentration is contributive to confirm the drug toxicity and describes drug exposure”; “Plasma concentrations and EEG provide essential cues regarding the diagnosis”.

The added paragraphs listed in question 3 are also in line with this comment.

5. Beta-lactam antibiotics are often combined with other antibiotics, such as aminoglycosides. Did the authors consider the possible neurotoxic effect of these antibiotics when administered with ceftriaxone?

We agree with the referee, other medications could provide CNS ADRs and thus could be confounding factors. Clinicians in pharmacovigilance centers provide a pharmacological, biological and chronological assessment and the use of other medications are always taken into account. Indeed other antibiotics such as other betalactams, metronidazole, or quinolones can provide CNS ADRs (Bhattacharyya and al, 2016; Deshayes and al, 2017). To better characterize our case reported and to answer the referee, we have added results of concomitant use of antibiotics (see 2.3 in the section results)

6. I consider it appropriate to add to the chapter Methods the criteria for adverse reactions that have been applied in the case of association with cefriaxone.

We agree with the referee. The section Material and methods has thus been revised in the manuscript:

-in the Materials and Methods section, population and cases: “Administration routes, daily dose (at the CNS ADR onset), concomitant use of other antibiotics and indications of ceftriaxone therapy were also collected.” (line 254)

-in the Materials and Methods section, statistical analysis: “We described general characteristics of serious case reports, as well as administration route, daily doses, concomitant use of other antibiotics, and indications.” (line 263)

References

-Bhattacharyya, S.; Darby, R.R.; Raibagkar, P.; Gonzalez Castro, L.N.; Berkowitz, A.L. Antibiotic-Associated Encephalopathy. Neurology 2016, 86, 963–971, doi:10.1212/WNL.0000000000002455.

-Deshayes, S.; Coquerel, A.; Verdon, R. Neurological Adverse Effects Attributable to β-Lactam Antibiotics: A Literature Review. Drug Safety 2017, 40, 1171–1198, doi:10.1007/s40264-017-0578-2.

-Singer, M.; Deutschman C.S.; Seymour, C.W.; and al. The Third International Consensus Definitions for Sepsis and Septic Shock (Sepsis-3). JAMA 2016, 315, 801-810, doi:10.1001/jama.2016.0287

Round 2

Reviewer 2 Report

I would like to thank the authors for considering my comments. The comments have been adequately resolved and I recommend the article for acceptance.